# Content Analysis on Sustainability Dimensions in DMOs' Social Media Videos Advertising the World's Most Visited Cities

**Mihai F. Băcilă** *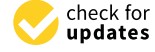**, Raluca Ciornea** *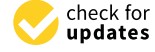**, Luiza M. Souca and Alexandra M. Drule**

Faculty of Economics and Business Administration, Babeș-Bolyai University, 400084 Cluj-Napoca, Romania
* Correspondence: mihai.bacila@econ.ubbcluj.ro (M.F.B.); raluca.ciornea@econ.ubbcluj.ro (R.C.)

**Abstract:** Rapid population expansion and poorly managed economic growth, unprecedented anthropogenic climate changes, non-renewable natural resources depletion, environmental pollution, social inequity, and loss of cultural integrity generate a global context that calls for urgent adoption of a sustainable development approach in major industries, including tourism. Sustainable tourism development requires the mobilization of tourism stakeholders at all levels and on the demand side through actions related to the travel decision-making process. To stimulate attitude formation and adoption of sustainable tourist behaviors, Destination Management Organizations (DMOs)—the main organizations responsible for sustainable destination development—need to adopt sustainable-oriented communication actions when building the destination image. As tourism stakeholders' perspective is under-investigated for destination image projection and communication, this paper aimed at assessing the integration of sustainable tourism principles in the promotion of destinations by DMOs; the focus was placed on video advertisements posted on the primary online source of tourism information, social media. Through a content analysis of DMOs' social media advertisements for the 50 most visited cities in the world, the current study revealed that elements covering all three dimensions of sustainability (economic, socio-cultural, natural) were featured in the commercial discourse, but not in a sustainable explicit standpoint. Besides, the content sporadically reflected sustainable governmental initiatives and projected responsible tourist behavior, while it lacked specific sustainable tourism-related terms. Moreover, several highly popular cities did not have promotional videos. The paper contributes to the body of knowledge on destination image formation by providing evidence from the supply's side along with an original content analysis grid which can be used to capture and evaluate the sustainable tourism dimensions as portrayed in advertising videos of cities. Additionally, it offers practical implications for DMOs' marketing communication strategies.

**Keywords:** destination image; sustainable tourism; projected destination image; DMO; city tourism; social media; content analysis



## 1. Introduction

Although the topic of sustainable tourism is not recent, the interest of academic, international governmental organizations, NGOs, and other stakeholders has expanded substantially over the past decade [1–3], accelerated by the increased general concern in sustainability. Despite this, research has shown that the advance in the theoretical discourse on sustainable tourism is associated with a slow-paced practical implementation [4], which is partially a consequence of the ongoing debates regarding its dimensions and the lack of a unique sustainable index/set of indices [5,6].

Inspired from by Brundtland Commission [7], the World Trade Organization [8] defines sustainable tourism as "tourism that takes full account of its current and future economic, social and environmental impacts, addressing the needs of visitors, the industry, the environment and host communities". Hence, focusing on the three-pillar interconnection

(environment, economy, society or socio-cultural) [2], sustainable tourism aims at meeting the current needs of tourists and host communities simultaneously with the protection and expansion of future opportunities [9,10]. Additionally, proper tourism management in a sustainable manner can raise awareness of the importance of preserving the environment, natural resources, and cultural heritage for future generations, along with enhancing tourists' experiences and satisfaction [11].

A significant and rapid sustainable tourism development implies mutually coordinated stakeholders [12] at all levels, from global to local. Over the past years, worldwide organizations and governments (e.g., United Nations, European Union) included in their agenda several sustainability development goals covering the tourism industry [13]. However, the main vectors in the sustainable development of tourist destinations, competition, and attracting tourists have been the organizations responsible for the management and marketing of tourist destinations, the DMOs [14,15]. Belonging partially or totally to governments and functioning at national, regional, or local levels [16], the DMOs go beyond engineering and coordinating the organizational aspects of travel destinations [17], representing their voice through marketing communication actions.

Given the intangibility characteristic of the tourism product associated with greater spatial and temporal distances between the intention formation-purchase-consumption stages, tourism is a highly information-based and information-intensive sector [18–20]. Tourists seek and use a wide range of information in the decision-making process by resorting to various sources, channels, and information technologies [21,22]. Hence, they are more informed, critical, and experimented, and increasingly involved in their tourism experiences and more concerned with engaging in responsible behavior when traveling [23–25]. With the rise in global awareness of the sustainability importance, tourists' attitudes towards over-tourism have begun to change, calling on the tourism industry to adopt sustainable approaches and provide more environmentally friendly and ethically correct products [26,27]. Consequently, the key stakeholders in the tourism industry and the DMOs need to correlate and harmonize their tourism marketing actions and strategies with sustainability issues, to project a destination image aligned with tourists' and society's growing preoccupation [28].

In this regard, and considering the limited investigation on sustainability depiction in communication actions [29,30], the present study aims at assessing the integration of the sustainable tourism concept into the projected image of tourist destinations. The paper reviews the case of promotion videos made by DMOs for several most visited cities in the world and shared on organizations' social media accounts, whilst the analysis covers the three pillars of sustainability, along with specific sub-components.

## 2. Literature Review

### 2.1. Tourism Sustainability

The tourism sector is undergoing a process of transformation in which the development and promotion of tourist destinations begin to follow a framework that embeds sustainability as a key driver to ensure long-term competitiveness [31–35]. The approach is suitable for all forms of tourism and destinations, from mass to niche tourism [36], although it is considerably challenging to apply in practice [6].

The multidimensional assessment of sustainable tourism is prevalent in the literature [37]. In their quest to identify sustainable tourism indicators, scholars have focused mostly on the so-called traditional dimensions derived from the concept of sustainability—economic, socio-cultural, and environmental/natural—while several have suggested new variables [5]. In many instances, the difference lies in the approach, as the traditional dimensions are broad enough to cover most of the new ones. By way of illustration, the environmental dimension can include the ecological aspects, the economic dimension can integrate the host community's well-being and so on, whereas tourist satisfaction is rather an overall outcome of sustainable tourism implementation [38–40]. If the traditional dimensions gained widespread academic recognition, there is no consensus

on the set of indices that make them [5], fueling the controversy revolving around the sustainable tourism concept [6].

The traditional view of sustainable tourism called for a balanced, integrated and harmonious development of the economic, socio-cultural and environmental dimensions [3,35,41–43], in such "a manner and at a scale" that can be maintained feasibly over the long term [44] (p. 29), by focusing on strategic solutions that maximize the tourist satisfaction and the benefits for local communities, while minimizing the adverse effects on the natural and cultural assets [45].

The regional or national economic development has a limited influence on the selection of a tourist destination (note that tourism is an important sector in the least developed countries too [46–48]), yet, several economic factors embodied by the destination image have an impact on tourist satisfaction and their intention to revisit and recommend the destination, in particular: tourist infrastructure quality and variety (e.g., choice of accommodation, selection of restaurants/dining places, entertainment/leisure/nightlife facilities, shopping facilities), the condition of the public infrastructure (e.g., health services infrastructure, internet and telecommunication infrastructure), accessibility and transportation system, price and value for money for goods and services [3,49–52]. In terms of economic sustainability, tourism can be seen as a commercial activity that drives economic prosperity and enhances the standard of living for the host communities. These positive effects are attributable to tourism's potential to develop facilities and infrastructure, stimulate business creation, investments, and export of local products, and serve as a direct or indirect source of employment and income [3,53–56]. This is the case for all categories of countries, regardless of the development status [57]. However, this dimension is not only about the economic gains and their fair distribution throughout the community but also about setting an optimal tourism growth pace and a threshold of growth that can be managed with minimal long-term adverse effects on the other pillars [5,58]. A poorly managed economic growth attributed to tourism can have economic downsides for the residents, including an unstable economic status consequence of a fluctuating income in destinations prone to tourism seasonality, deterioration in housing affordability and gentrification, inflated prices for goods and services, and an increased cost of living [3,59–61], which in conjunction with other socio-cultural and environmental negative consequences can trigger tourismphobia [60,62].

Tourist experience can be enhanced by visiting historical and cultural sites, museums, and local markets, enjoying local cuisine and products (ethnic restaurants, street food stands), watching traditional artistic activities (music, dance, crafts, festivals), sleeping in traditional accommodation and interacting with residents which provide opportunities to actively learn about their way of life [63–65]. Previous research showed that cultural heritage contributes to destination competitiveness [66,67], a high-quality cultural experience has a positive influence on satisfaction, intention to revisit and recommend the destination [63,68,69], while a positive social-related tourism experience increases the intention to recommend the destination [68]. From the sustainable social perspective, enjoying authentic tourist experiences linked to cultural heritage, traditions, artifacts, architecture, and local history must be done without triggering or with a limited negative impact [70]. Thus, tourism must ensure social inclusion and cohesion, inter- and intra-generational equity, human capital development, the preservation of social diversity, traditional values, and local identity [42,53,71,72]. Along with the key stakeholders engaged in the socio-cultural facets of tourism, the locally owned and controlled micro-businesses aimed at tourists generate income and raise the standard of living and the quality of life for the host community [73–75]. However, in the absence of an integrated socio-cultural management, the local stakeholders' actions can result in loss of cultural integrity, through acculturation, hybridization of culture, cultural commodification [76]; these add to overcrowding and detrimental influence on attractions/sites/museums/events, and impediment of routine activities of residents [59,77].

The existence of a well-preserved/authentic natural environment, natural attractions or recreational green spaces can yield a competitive advantage for a destination, drawing tourists [3,66,78,79]. Visiting green areas and connecting with nature lead to a better and unique tourist experience [79,80], which reflects on satisfaction [49,81], intention to revisit and recommend the destination [45,57], and also increases tourists' environmental awareness [80]. Natural sustainability requires that the destinations development and meeting tourists' needs must ensure a healthy natural environment, achieved by protecting the natural areas, restoring and proper management of ecosystems, biodiversity conservation, reduced pollution of land, air, water, and sustainable use of land resources [5,42,72,73,82]. To ensure the optimal use of the environmental resources with preservation of natural heritage and biodiversity, the host community needs to be involved [73], educated, and encouraged to use ecological engineering and architecture (e.g., green buildings), resource-efficient technologies (e.g., renewable energy) and to provide eco-friendly alternatives for tourists (e.g., eco-friendly/greener transportation and activities) [79,83,84]. Simultaneously, the main local stakeholders need to consider the mass-tourism and over-tourism's negative impact on the environment [85] and cultural heritage, and to adopt strategies to counteract it [79,86]. If not properly managed, tourism development can have harmful effects such as land, air, water, noise, and light pollution, overloaded infrastructure, biodiversity and ecosystem loss and degradation, and improper/illegal waste disposal [59,73,79,87,88]. These affect not only the local community, nature, and wildlife, but can also discourage tourism [89,90].

*2.2. Destination Image*

Destination image is one of the topics frequently covered in tourism academic research [91] and refers to the global impression [92] or "the sum of beliefs, ideas and impressions that a person has" [93] (p. 18) of a destination. The image formation mental process begins with the projection of an organic image solely grounded on the consumer's general knowledge about the destination, without external commercial exposure [94–96]. In the second phase, the organic image is converted into an induced image under the influence of commercial materials [94–96]. Hence, the induced image is the result of deliberate promotional efforts of tourism organizations such as DMOs.

Despite the general view on the multidimensionality of destination image [97], there is no consensus on the components that make its identity. The research framework distinguishes between cognitive, affective, and conative evaluations, holistic and individual attribute-based perceptions, functional and psychological characteristics [91,92,95,98–101]. According to the extant literature, the construct covers a wide spectrum of dimensions, such as tourist facilities and leisure activities, natural and cultural attractions, economic development and political stability, reputation, safety, atmosphere, interaction with locals, entertainment, events, and nightlife [3,95,102]. Regardless of the multi-indicators approach, to serve the sustainable tourism goals, the projected destination image must encompass, although not exclusively, the traditional interrelated dimensions: economic, socio-cultural and environmental.

This mental representation of the tourist destination plays a prominent role in all stages of the consumer's decision-making process, influencing the pre-visiting selection, the on-site experience [69,103,104], the post-visiting satisfaction [3,49,51,105–109] and the future behavioral intentions such as the intention to recommend [106–109] and to revisit the destination [106–111]. Additionally, previous research confirmed the positive impact of destination image on the intention to select or visit a sustainable destination [112–114], respectively on the tourists' intention to adopt a sustainable behavior [115].

In today's intense competition amplified by the fast-growing pace of tourism [102] and considering the influence of the destination image, an in-depth understanding of its formation process is needed [116] along with the identification of its attributes [117]. Besides, it is of utmost importance to undertake a more frequent evaluation and enhancement

of the destination competitiveness [102] by placing the image dimensions at the core of marketing and management activities, such as differentiation [118] and promotion [105].

### 2.3. Destination Image and DMOs Communication on Social Media

The destination image influence in the travel decision-making process starts with the pre-visiting selection stage [104]. Considering that tourism-related services are high-priced and cannot be assessed prior to consumption, they are expected to lead to high-risk high-involvement purchases [101,119]. Consequently, tourists' time and energy will be reflected in a rather complex buying behavior [120], where the information collection (quality, amount, and sources of information) and review play a critical part [101,119,120]. In this information-intensive industry, the traditional information sources have been substituted over the past two decades by the emerging and proliferation of ICTs (information and communication technologies), travel websites and social media platforms, revolutionizing the tourists' travel planning process [101,120–124]. As social media have developed into the most abundant, diverse, interactive, and powerful source of online information, exhibiting a higher level of reliability [100,120,124,125] than the fully paid marketing communication channels [101], tourists widely use it in the pre-visiting selection stage, to search for destinations, accommodation, dining places and other facilities, leisure and entertainment activities, transportation [100,101,104,119,120,123,124,126,127]. Furthermore, extant research provides some evidence to support the influence of social media information and information source on the formation process of cognitive, affective, and overall destination image, with an effect on destination selection [100,101,104,123].

The communication efforts of DMOs and other tourism stakeholders play a pivotal role in the process of destination image formation [95,96,128]. The worldwide dominance of ICTs, internet technology, and the rise in popularity of online information sources among tourists, also reflect the highly competitive tourism business environment [124,125], changing how DMOs and other tourism stakeholders create and implement their communication marketing strategies [100,119,120]. Inevitably, they pushed the DMOs and other organizations to shift a significant amount of marketing communication activities from the traditional media towards online, and to resort to digital channels such as own websites and multiple Social Media platforms (Facebook, Instagram, Twitter, Weibo, YouTube) when building the destination image and destination-brand identity [100,101,104,119]. Emerged as a low-cost, flexible, efficient, and global reach marketing tool [119,121,124], social media became a prominent element of DMOs' marketing strategies used, along their official websites, to heighten the destination competitiveness, the projected image, and the intention to visit [101,104,123,129]. However, despite social media providing the means to enable interactivity and bidirectional communication, encouraging tourists' engagement, collaboration and eWOM, additionally to other specific management and marketing functions [104,119,121,129], a significant number of DMOs limit its use as a traditional marketing tool, in the sense of dissemination of information [119]. For instance, a study on European DMOs empirically confirmed that the content posted on Facebook, Instagram, Twitter, and YouTube was predominantly promotion-related, covering the destination attractions, accommodation, restaurants, events, and the official tourism website [119]. Additionally, in other studies, the tourism supply-side was found deficient in building dialogues and creating interactions with travelers on social media [119], although DMOs' information richness and efforts on social media have the potential to stimulate tourists' engagement on the platforms [121]. In view of previous judgements, some authors place the DMOs' social media efforts in the experimental stage [119,130]. When it comes to communication content, DMOs and the other tourism stakeholders accountable for the sustainable development of destination image [104], must consider that social media also provides opportunities and means to adopt marketing communication strategies with a more sustainable approach [131,132]. Specifically, when projecting the social media communication strategies, DMOs need to encompass the sustainability principles in their advertisement and messages, as it can enhance tourists' attitude towards a sustainable destination, the

creation of a sustainable destination image and the adoption of responsible tourist behavior [104,131,133,134]. Regarding the format of online marketing communication, tourism stakeholders (including DMOs) increasingly use destination promotion videos, considering them as powerful communication tools that can create cognitive, affective, and conative responses during the tourist information process [135,136].

The destination image's influence extends to the other stages of the travel decision-making process, namely the on-site experience, the post-visiting evaluation (level of satisfaction) and the intentions to revisit and recommend [104–106,124]. The subsequent evaluations made by tourists imply a comparison between the expectations built on the pre-visiting destination image and the actual performance quality and perceived experience of the trip [51,103,137]. For that reason, DMOs must be consistent in the image projection [104] and avoid generating unrealistic expectations by exaggerating the destination's characteristics in the communication materials [57]. This can be of utmost importance when considering that the proliferation of ICTs and the online environment makes it possible to anyone in the world to easily generate content–UGC [123,138]. Particularly, it encourages tourists to share their on-site or post-visiting travel experiences [104,121] through UGC, such as reviews, comments, photos, and videos [119,139,140], predominantly created on social media platforms [100,101,123]. In this context, tourists actively participate in the co-creation of the overall destination image, ergo affecting other visitors' decision-making process and concomitantly weakening the DMOs' control, influence, and leading role as information source [100,101,104,123,138,140]. Moreover, there have been few attempts to empirically compare the destination image projected by DMOs and the one perceived by tourists as revealed through UGC; the results confirm that congruence between the expectations created by DMOs and the reality and travel experience might increase the brand credibility, tourist satisfaction, future intentions to revisit and recommend the destination, and hence the competitive advantage, while the reverse is true for discrepancies [123,140].

To develop efficient marketing strategies, tourism stakeholders need to understand the role of behavioral drivers [141] in the travel decision-making mechanism; hence, not surprisingly, the subject has triggered a large body of tourism literature. As a result, a significant number of scholars emphasized the importance of an information source as an antecedent for destination image formation, and have extensively explored this relationship [101,104,142]. However, tourists retrieve information from multiple sources, simultaneously [101,120,138], and the influence of various sources is still under-investigated [100]. In particular, despite the fact that increased social-media-based content (DMOs-, tour operators- and travel agencies-, media- and user/tourist- generated) supports the robust effect of social media information sources on the destination image formation, the topic has not been thoroughly examined in an empirical manner [100,101,104]. With Facebook and Twitter in the spotlight [119], extant research has addressed tourists' perspectives in matters such as social media usage and its influence on behavior, but, as formerly stated, without an in-depth and holistic understanding of the relationship between social media information and the formation process of destination image [101,120,143] or sustainable destination image [104]. Instead, the tourism stakeholders' perspective was scarcely documented; hence, little is known about DMO and other organizations' content creation and communication strategies on social media, both in general and in the context of tourist destination image creation [104,119,121]. Additionally, an overview of the literature on tourism marketing communication, unveils that relatively little insight has been put into the tourism advertisements and the sustainability dimensions in these advertisements, regardless of traditional media or social media distribution [29,30,104].

Following the lines of argumentation above, and considering the exploratory nature of the study, we propose the following research questions:

RQ1: If, how, and to what extent the dimensions of sustainable tourism (environment, socio-cultural, economic) appear in the advertisements.

RQ2: If and to what extent the governments/DMOs' initiatives for sustainable tourism development (on the 3 dimensions) show up in the advertisements.

RQ3: If and what sustainable tourist behavior is displayed in the advertisements.

## 3. Methodology

To assess the way that sustainability dimensions are reflected in DMOs' social media video advertisements, content analysis has been employed. The method is extensively used in communication and mass-media content studies [144,145], yet it has been described as a relatively novel approach when it comes to examining the destination image projection [123]. Given the qualitative nature of the investigation and the possible drawbacks implied, special attention has been paid to the research design for the purpose of ensuring appropriate objectivity, systematization, sampling procedure, and reliability [144,146,147].

To secure systematization, a literature review on sustainable destination image projection and a focus group between researchers were carried out to establish the code names, categories, examples, and operational definitions [144,146]. The final grid included a list of 70 items, divided by destination perspective, sustainable tourism dimensions (environmental, economic, socio-cultural), government initiatives for sustainable development related to environment and tourism, the explicit appearance of tourists in the video, and sustainable tourism-related terms (see Table A1 in Appendix A). The items were dichotomized, and coded as absent (0) or present (1) [148].

Objectivity was enhanced through detailed procedures, training coders, and measure pretesting [144]. Following the literature recommendation, the research team selected and trained two coders to increase their familiarity with the content analysis technique and the coding scheme [144,149]. Additionally, to avoid ambiguity, they were provided with a codebook with code names, rules and procedures, categories, operational definitions, and examples [150,151]. The two coders reviewed the coding manual, and discussed the coding rules and the definitions [152]. Further, the measure pretesting implied observers' autonomous judgement [144] of 10 similar videos that were not comprised in the sample. The inter-rater reliability among the coders has been assessed with Cohen's kappa statistic, a metric commonly used to measure the agreement between two raters on nominal scales [153]. Although the Cohen's kappa index value was 0.924 ($p < 0.001$), the differences were debated and jointly reviewed until a consensus was reached [152]. Once reliability has been established in pretesting, the grid has been applied independently, by each observer, to each video in the sample. Cohen's kappa coefficients confirmed the reliability of the coding process, with an almost perfect intercoder agreement (kappa = 0.947, $p < 0.001$) in the case of tourism satisfaction and perfect agreement for the other categories (kappa = 1.000, $p < 0.001$) [149].

Taking into account the methodology and the nature of the topic, it was opted for the purposing sampling technique, a method which allows for deliberately choosing the sample units and is highly used in the content analysis [152]. The sample included the 50 most visited cities in the world by foreign tourists [154] for which the correspondent DMOs have posted communication/presentation videos on their official YouTube and Facebook pages. Considering the latest destination advertising video uploaded on the DMOs' social media accounts, the sample covered: Bangkok, London, Macao, Singapore, Paris, Dubai, New York City, Kuala Lumpur, Istanbul, Delhi, Antalya, Mumbai, Phuket, Rome, Tokyo, Pattaya, Taipei, Guangzhou, Prague, Seoul, Amsterdam, Miami, Osaka, Los Angeles, Shanghai, Ho Chi Minh City, Denpasar, Barcelona, Las Vegas, Milan, Chennai, Vienna, Johor Bahru, Jaipur, Cancun, Berlin, Cairo, Athens, Orlando, Moscow, Venice, Madrid, Ha Long, Dublin, Hanoi, Toronto, Johannesburg, Sydney, Munich and Jakarta. Despite being highly popular destinations, several cities (Shenzhen, Taipei, Mecca, Medina, Agra, Riyadh, and Florence) were excluded from the list because they lacked a video advertisement. Based on the designated tourism regions, the sample comprised: 23 cities in Asia and the Pacific, 17 cities in Europe, 7 cities in the Americas, and 3 in the Middle East and Africa region. Although there is no common ground in setting up the sample size for qualitative research [152], there are scholars suggesting that a sample over 30 units can be difficult to analyze in the case of interviews [155], whilst a sample size of 50–200 units is manageable in case of

advertisements [156]. Hence, the sample size of the actual investigation meets the proposed requirements, along with the multiples of 10 "rule" similar to other studies found in the qualitative research literature [155]. The total video material was 130 min and 32 s long, covering the years 2012–2020, a time interval before WHO declared the COVID-19 outbreak a global pandemic. The unit of observation was the video, and the unit of analysis was the scene.

Data analysis consisted of relative frequencies distribution for items examination and the Chi-square (Pearson's Test for Independence) test for regional comparisons (similar approach in [157]), carried out with Microsoft Excel and IBM SPSS 20 software.

## 4. Results

The terminology used in the verbal/text advertising discourse can provide explicit evidence of a sustainable approach in communication. Despite that, none of the videos in the analysis included specific sustainable tourism-related terms such as: sustainability, sustainable development/tourism, equitable/responsible/eco/green tourism, eco-friendliness, nature/biodiversity/eco-system preservation, cultural preservation, inclusivity, fair/equitable income distribution, poverty alleviation, responsible travel/behavior.

Of particular interest in communicating the commitment to sustainability are the major facets of the destination image projected in the commercial discourse (Table 1). Nearly all the videos featured the urban destination side (92%), while considerably fewer displayed the natural (natural areas with mountains or seas 48%, and/or preserved natural attractions 28%) and the cultural (historical buildings or ruins 34%, and/or cultural heritage 38%) facets.

**Table 1.** Facets of projected destination image.

| Items | Asia/Pacific | | Europe | | Americas | | Middle E./Africa | | Total | |
|---|---|---|---|---|---|---|---|---|---|---|
| | **n** | **%** | **n** | **%** | **n** | **%** | **n** | **%** | **n** | **%** |
| Natural areas destinations-sea/mountains | 12 | 52.2 | 5 | 29.4 | 6 | 85.7 | 1 | 33.3 | 24 | 48.0 |
| Preserved nature attractions-secular forests/wild areas | 6 | 26.1 | 4 | 23.5 | 3 | 42.9 | 1 | 33.3 | 14 | 28.0 |
| Urban areas | 20 | 87.0 | 16 | 94.1 | 7 | 100 | 3 | 100 | 46 | 92.0 |
| Cultural destinations-historic city/ruins | 6 | 26.1 | 10 | 58.8 | 0 | 0.0 | 1 | 33.3 | 17 | 34.0 |
| Cultural destinations-cultural heritage | 12 | 52.2 | 4 | 23.5 | 2 | 28.6 | 1 | 33.3 | 19 | 38.0 |

Chi-square statistical analyses revealed significant differences by regions when promoting the historic landscape ($\chi^2$ = 8.917, df = 3, *p* = 0.030). In-depth insights detected the use of frames with historical areas/ruins in European (58.8%), Middle East/African (33.3%) and Asian/Pacific (26.1%) tourism videos, whereas they were absent in American tourism advertisements. Additionally, a rank-frequency distribution placed the American tourism videos first for the natural exposure (mountains, seas, secular forests, wilderness), the European advertisements for the historic landscape and the Asian/Pacific ones for the cultural heritage.

The answers to the research questions were addressed in the following sub-sections.

## 4.1. Evidence on Sustainable Tourism Dimensions–RQ1

To project a sustainable destination image that mirrors the sustainable tourism development in a region, DMOs' advertising discourse needs to encompass elements associated with all three pillars of sustainability: natural/environmental, socio-cultural, and economic.

The natural/environmental dimension of sustainable tourism (Table 2) was largely depicted through frames featuring clean environment (96%), parks and green spaces in the city (90%), mild or pleasant weather (86%), natural attractions such as the sea, mountains, or rivers (80%), tourists taking strolls in parks or natural areas (48%), natural areas outside the city (44%). In spite of having secondary importance for tourists, two other positive environmental aspects had a widespread presence among the advertisements: the lack of crowds (84%) and reduced traffic (70%). On the contrary, nature sounds were used as ambient sounds only in the European (17.6%) and Asian/Pacific (17.4%) videos.

**Table 2.** Sustainable tourism–Natural/environmental dimension.

| Items | Asia /Pacific | | Europe | | Americas | | Middle E./Africa | | Total | |
|---|---|---|---|---|---|---|---|---|---|---|
| | **n** | **%** | **n** | **%** | **n** | **%** | **n** | **%** | **n** | **%** |
| **Natural/environmental dimension** | | | | | | | | | | |
| Natural attractions–sea, mountain, river | 18 | 78.3 | 13 | 76.5 | 7 | 100 | 2 | 66.7 | 40 | 80.0 |
| Natural areas outside the city | 13 | 56.5 | 4 | 23.5 | 4 | 57.1 | 1 | 33.3 | 22 | 44.0 |
| Parks and green spaces in the city | 20 | 87.0 | 16 | 94.1 | 7 | 100 | 2 | 66.7 | 45 | 90.0 |
| Mild/nice weather | 22 | 95.7 | 12 | 70.6 | 6 | 85.7 | 3 | 100 | 43 | 86.0 |
| Clean environment | 22 | 95.7 | 17 | 100 | 7 | 100 | 2 | 66.7 | 48 | 96.0 |
| Reduced traffic | 15 | 65.2 | 14 | 82.5 | 4 | 57.1 | 2 | 66.7 | 35 | 70.0 |
| Lack of crowds | 19 | 82.6 | 15 | 88.2 | 6 | 85.7 | 2 | 66.7 | 42 | 84.0 |
| Natural ambient sound | 4 | 17.4 | 3 | 17.6 | 0 | 0.0 | 0 | 0.0 | 7 | 14.0 |
| Outdoor natural strolls/trekking | 12 | 52.2 | 6 | 35.3 | 5 | 71.4 | 1 | 33.3 | 24 | 48.0 |
| **Nature harming activities and effects** | | | | | | | | | | |
| Intense traffic | 10 | 43.5 | 6 | 35.3 | 2 | 28.6 | 2 | 66.7 | 20 | 40.0 |
| Crowds | 7 | 30.4 | 4 | 23.5 | 2 | 28.6 | 1 | 33.3 | 14 | 28.0 |
| Implied water and air pollution | 3 | 13.0 | 2 | 11.8 | 0 | 0.0 | 1 | 33.3 | 6 | 12.0 |
| Implied sound pollution | 5 | 21.7 | 2 | 11.8 | 2 | 28.6 | 0 | 0.0 | 9 | 18.0 |
| Intense public lightning | 13 | 56.5 | 6 | 35.3 | 3 | 42.9 | 1 | 33.3 | 23 | 46.0 |

The results of a rank-frequency distribution confirmed the previous findings on American advertisements' dominance for natural exposure, as the videos recorded the highest frequency of frames for natural attractions, natural areas outside the city, parks in the city, clean environment, and natural outdoor strolls. In addition, the European tourism videos were placed first for exhibiting a clean environment, reduced traffic, and lack of crowds, whereas Middle East/African advertisements got the same rank for the mild weather.

On top of that, the videos showcased several nature harming activities and effects, such as footage of intense traffic (40%), crowds (28%), intense public lightning (48%), implied sound pollution (18%), implied water and air pollution (12%). Based on the rank-frequency distribution of the negative natural features, the Middle East/African tourism videos were placed first for exhibiting intense traffic, crowds, and implied water/air pollution, the American advertisements for implied sound pollution and the Asian/Pacific ones for intense public lighting.

Chi-square analyses revealed no statistically significant differences between regions when displaying the environmental items in tourism videos ($p > 0.050$).

The socio-cultural dimension of sustainable tourism (Table 3) was depicted through footage including scenes with cultural attractions (90%), preserved historical buildings (88%) and museums, theatres, or art galleries (68%). Only 4% of the videos, all Asian/Pacific, highlighted having received a UNESCO award or distinction. The Chi-square statistical analyses revealed a significant association between regions and the incorporation of museums/theaters/art exhibitions ($\chi^2 = 10.082$, df = 3, $p = 0.018$), with the Asian and European advertisements more likely to dominate the display.

In terms of socio-cultural heritage preservation, the results showed that the tourism advertisements included frames featuring instances of culture preservation (84%), traditional food and scenes from the local community's way of life (76%), local traditions and cultural identity (60%), local crafts and handmade art (52%), at least one festival or event (46%), dancing and music (44%), and healthy local products (26%). Even though local language and traditional music are elements of a culture, less than half of the tourism videos (40%) included them in the soundtrack to enhance the authentic experience, the only regional exception being the Asian/Pacific advertisements.

Chi-square analyses confirmed inter-regional statistically significant differences for the items covering the display of local crafts and handmade arts ($\chi^2 = 16.516$, df = 3, $p = 0.001$), traditional food ($\chi^2 = 10.418$, df = 3, $p = 0.015$), cultural preservation ($\chi^2 = 14.486$, df = 3, $p = 0.002$), scenes from the local community way of life ($\chi^2 = 8.015$, df = 3, $p = 0.046$), traditional music or local language as ambient sound ($\chi^2 = 8.080$, df = 3, $p = 0.044$). Particularly, the Asian/Pacific tourism videos were more likely to present frames that include crafts/handmade arts, and ambient sounds, whereas the Middle East/African and Asian advertisements are more likely to display scenes unveiling cultural preservation, traditional food, and local community way of life. In contrast, these cultural items are less likely to be depicted in the American videos.

Additional insights on social-cultural activities that include tourists showed a frequent display in tourism advertisements for the interactions with the locals (86%), visits to cultural or historical attractions (80%), and cultural experiences such as attending opera, theatre performances or art exhibitions (64%), consumption of traditional products (62%), culinary experiences at traditional (60%) or non-traditional restaurants (56%). Tourists were shown less often while listening to traditional music or watching local dances (44%), or visiting rural areas/villages and traditional houses (24%). Instead, many videos feature friendly local people (82%).

The inter-regional Chi-square comparisons revealed several significant differences for tourism advertisements. Specifically, in contrast to American and Middle East/African advertisements, the European and Asian/Pacific tourism video included more frequently frames that depict visits to cultural and historical attractions ($\chi^2 = 12.932$, df = 3, $p = 0.005$) or attendances at opera and theatre performances ($\chi^2 = 7.993$, df = 3, $p = 0.046$). Moreover, the Asian/Pacific and Middle East/African advertisements were more likely to present tourists consuming traditional products ($\chi^2 = 9.015$, df = 3, $p = 0.028$).

**Table 3.** Sustainable tourism–Socio-cultural dimension.

| Items | Asia/Pacific | | Europe | | Americas | | Middle E./Africa | | Total | |
|---|---|---|---|---|---|---|---|---|---|---|
| | n | % | n | % | n | % | n | % | n | % |
| **Cultural and historical assets** | | | | | | | | | | |
| Cultural attractions | 22 | 95.7 | 17 | 100 | 5 | 71.4 | 1 | 33.3 | 45 | 90.0 |
| Preserved historical buildings | 20 | 87.0 | 16 | 94.1 | 6 | 85.7 | 2 | 66.7 | 44 | 88.0 |
| UNESCO awards/distinctions | 2 | 8.7 | 0 | 0.0 | 0 | 0.0 | 0 | 0.0 | 2 | 4.0 |
| Museums, theatres, art galleries | 18 | 78.3 | 13 | 76.5 | 3 | 42.9 | 0 | 0.0 | 34 | 68.0 |
| **Socio-cultural heritage preservation** | | | | | | | | | | |
| Festival/event | 10 | 43.5 | 10 | 58.8 | 3 | 42.9 | 0 | 0.0 | 23 | 46.0 |
| Crafts/handmade art | 19 | 82.6 | 7 | 41.2 | 2 | 28.6 | 1 | 33.3 | 26 | 52.0 |
| Local traditions and cultural identity-including folk art and clothing | 18 | 78.3 | 7 | 41.2 | 3 | 42.9 | 2 | 66.7 | 30 | 60.0 |
| Dance/music | 12 | 52.2 | 7 | 41.2 | 2 | 28.6 | 1 | 33.3 | 22 | 44.0 |
| Traditional food | 22 | 95.7 | 11 | 64.7 | 3 | 42.9 | 2 | 66.7 | 38 | 76.0 |
| Green label/healthy products | 8 | 34.8 | 5 | 29.4 | 0 | 0.0 | 0 | 0.0 | 13 | 26.0 |
| Cultural preservation | 23 | 100 | 13 | 76.5 | 3 | 42.9 | 3 | 100 | 42 | 84.0 |
| Scenes from the local community's way of life | 21 | 91.3 | 10 | 58.8 | 4 | 57.1 | 3 | 100 | 38 | 76.0 |
| Ambient sound–traditional music/local language | 14 | 60.9 | 4 | 23.5 | 1 | 14.3 | 1 | 33.3 | 20 | 40.0 |
| **Socio-cultural activities** | | | | | | | | | | |
| Activities–listening to music, dancing | 12 | 52.2 | 7 | 41.2 | 2 | 28.6 | 1 | 33.3 | 22 | 44.0 |
| Activities–visits to cultural/historical attractions | 20 | 87.0 | 16 | 94.1 | 3 | 42.9 | 1 | 33.3 | 40 | 80.0 |
| Activities–attending opera or theatre performances or art exhibitions | 17 | 73.9 | 12 | 70.6 | 3 | 42.8 | 0 | 0.0 | 32 | 64.0 |
| Activities–interaction with local people | 22 | 95.7 | 12 | 70.6 | 6 | 85.7 | 3 | 100 | 43 | 86.0 |
| Activities–visits to rural areas/villages/traditional houses | 5 | 21.7 | 4 | 23.5 | 2 | 28.6 | 1 | 33.3 | 12 | 24.0 |
| Activities–consuming traditional products | 19 | 82.6 | 8 | 47.1 | 2 | 28.6 | 2 | 66.7 | 31 | 62.0 |
| Culinary experiences in a traditional restaurant | 17 | 73.9 | 9 | 52.9 | 2 | 28.6 | 2 | 66.7 | 30 | 60.0 |
| Culinary experiences in a non-traditional restaurant | 14 | 60.9 | 9 | 52.9 | 4 | 57.1 | 1 | 33.3 | 28 | 56.0 |
| Friendly locals | 20 | 87.0 | 13 | 76.5 | 6 | 85.7 | 2 | 66.7 | 41 | 82.0 |

To capture the economic dimension of sustainable tourism (Table 4), the analyzed tourism videos included scenes depicting local businesses (84%), nightlife (74%), a good transport system (72%), shopping as a leisure activity (64%), good infrastructure (58%), a strong economy (50%), and hotels/ resorts (40%). Despite the increasing association with sustainable development, the smart-city solutions were hardly reflected in advertisements

(14%), being present only in Asian/Pacific and European videos. Furthermore, all urban destinations were promoted as being safe for travelers and locals alike, avoiding the display of frames hinting at underdeveloped or high-crime areas/neighborhoods.

**Table 4.** Sustainable tourism–Economic dimension.

| Items | Asia/Pacific | | Europe | | Americas | | Middle E./Africa | | Total | |
|---|---|---|---|---|---|---|---|---|---|---|
| | **n** | **%** | **n** | **%** | **n** | **%** | **n** | **%** | **n** | **%** |
| **Economic dimension** | | | | | | | | | | |
| Strong economy | 13 | 56.5 | 7 | 41.2 | 3 | 42.9 | 2 | 66.7 | 25 | 50 |
| Safe cities-the absence of underdeveloped /high-crime areas/neighborhoods | 23 | 100 | 17 | 100 | 7 | 100 | 3 | 100 | 50 | 100 |
| Good infrastructure | 13 | 56.5 | 10 | 58.8 | 4 | 57.1 | 2 | 66.7 | 29 | 58.0 |
| Smart city | 5 | 21.7 | 2 | 11.8 | 0 | 0.0 | 0 | 0.0 | 7 | 14.0 |
| Good transport system | 17 | 73.9 | 14 | 82.4 | 3 | 42.9 | 2 | 66.7 | 36 | 72.0 |
| Hotels/resorts | 11 | 47.8 | 2 | 11.8 | 5 | 71.4 | 2 | 66.7 | 20 | 40.0 |
| Local businesses | 21 | 91.3 | 14 | 82.4 | 5 | 71.4 | 2 | 66.7 | 42 | 84.0 |
| Activities–shopping | 16 | 69.6 | 9 | 52.9 | 5 | 71.4 | 2 | 66.7 | 32 | 64.0 |
| Nightlife | 16 | 69.6 | 13 | 76.5 | 6 | 85.7 | 2 | 66.7 | 37 | 74.0 |

The inter-regional Chi-square comparisons revealed significant differences for tourism advertisements displaying frames featuring hotels/resorts, but not for the other economic items ($p > 0.050$). Hence, the American and Middle East/African advertisements were more likely to have scenes with tourist accommodations such as hotels and resorts ($\chi^2 = 10.004$, df = 3, $p = 0.019$).

*4.2. Government Initiatives Related to Sustainable Tourism Development–RQ2*

Voicing, directly or indirectly, the government's commitment to sustainability and sustainable tourism development gives more weight and credibility to the sustainability-related matters conveyed in the commercial discourse of a destination image (Table 5).

The government initiatives directed at the environmental dimension of sustainable tourism were mainly depicted through frames featuring electric transportation such as trains, trams, and the metro (48%), biodiversity preservation (plants–44% and/or animals–44%) and focus on natural resources (30%). Less popular was the footage including green buildings (18%), whilst the renewable energy sources (2%) were seen only in the European tourism videos.

Chi-square statistical analyses showed inter-regional significant differences for the items covering the usage of electric transport vehicles and the focus on natural resources. In particular, the American and Asian/Pacific tourism videos were more likely to present frames focusing on natural resources ($\chi^2 = 8.093$, df = 3, $p = 0.044$), while the electric transport vehicles are more likely to be depicted in the European advertisements ($\chi^2 = 8.448$, df = 3, $p = 0.038$).

In terms of socio-cultural government initiatives, the pursuit of sustainable tourism development was primarily presented through frames implying the idea of tolerance and understanding of gender, sexuality, race, and age diversity (84%). Besides, a considerable number of videos reflected the respect towards local food producers (76%) and the promotion of traditional/eco-friendly street food or food markets (56%).

**Table 5.** Government initiatives related to sustainable tourism development.

| Items | Asia/Pacific | | Europe | | Americas | | Middle E./Africa | | Total | |
|---|---|---|---|---|---|---|---|---|---|---|
| | **n** | **%** | **n** | **%** | **n** | **%** | **n** | **%** | **n** | **%** |
| **Natural/environmental dimension** | | | | | | | | | | |
| Focus on natural resources | 9 | 39.1 | 1 | 5.9 | 4 | 57.1 | 1 | 33.3 | 15 | 30.0 |
| Biodiversity preservation–plants | 13 | 56.5 | 4 | 23.5 | 4 | 57.1 | 1 | 33.3 | 22 | 44.0 |
| Biodiversity preservation–animals | 13 | 56.5 | 4 | 23.5 | 4 | 57.1 | 1 | 33.3 | 22 | 44.0 |
| Renewable energy | 0 | 0.0 | 1 | 5.9 | 0 | 0.0 | 0 | 0.0 | 1 | 2.0 |
| Green buildings | 5 | 21.7 | 4 | 23.5 | 0 | 0.0 | 0 | 0.0 | 9 | 18.0 |
| Electric transport vehicles: train, tram, metro | 8 | 34.8 | 13 | 76.5 | 2 | 28.6 | 1 | 33.3 | 24 | 48.0 |
| **Socio-cultural dimension** | | | | | | | | | | |
| Respect for the local food producers | 21 | 91.3 | 11 | 64.7 | 4 | 57.1 | 2 | 66.7 | 38 | 76.0 |
| Promoting traditional/eco-friendly food-local food markets/street food | 18 | 78.3 | 7 | 41.2 | 2 | 28.6 | 1 | 33.3 | 28 | 56.0 |
| Tolerance and understanding–gender, sexuality, race and age diversity | 21 | 91.3 | 13 | 76.5 | 6 | 85.7 | 2 | 66.7 | 42 | 84.0 |
| **Economic dimension** | | | | | | | | | | |
| Traditional accommodation units | 1 | 4.3 | 1 | 5.9 | 1 | 14.3 | 0 | 0.0 | 3 | 6.0 |
| Small local businesses owned by diverse locals | 17 | 73.9 | 10 | 58.8 | 4 | 57.1 | 1 | 33.3 | 32 | 64.0 |
| Service quality–friendly sellers | 20 | 87.0 | 12 | 70.6 | 4 | 57.1 | 3 | 100 | 39 | 78.0 |
| Service quality–small physical premises | 21 | 91.3 | 15 | 88.2 | 5 | 71.4 | 2 | 66.7 | 43 | 86.0 |
| Local partnerships-info centres and local tourism associations | 1 | 4.3 | 4 | 23.5 | 0 | 0.0 | 0 | 0.0 | 5 | 10.0 |

The inter-regional Chi-square comparisons confirmed significant differences for tourism advertisements promoting local street food, but not in the case of diversity or respect for local food producers ($p > 0.050$). More precisely, the Asian/Pacific tourism advertisements followed at a distance by the European ones, were more likely to include frames with local street cuisine sold in local food markets or as street food ($\chi^2 = 8.905$, df = 3, $p = 0.031$).

The economic government initiatives anchored on sustainable tourism were primarily depicted through footage exposing the service quality of locally owned micro-business (86%) and friendly sellers (78%), and the diversity in ownership for small local tourism-related businesses (64%). Despite being important for tourists, the advertising discourse rarely incorporated frames with traditional accommodation units (6%) or local partnerships as info centers and local tourism associations (10%), the latter being present only in European and Asian/Pacific advertisements.

Chi-square statistical analyses revealed no significant differences by region when displaying the governmental economic initiatives in tourism advertisements ($p > 0.050$).

### 4.3. Sustainable Tourist Behavior Displayed in Advertisements–RQ3

The explicit presence of tourists in the advertising discourse can be used as a part of education to induce sustainable behaviors expected while visiting the destination. Besides, if presented while joyfully experiencing tourist activities, it not only enhances the destination image but also can promote the idea that sustainable tourism development is done by meeting tourists' needs and ensuring their satisfaction (Table 6).

**Table 6.** Explicit presence of tourists.

| Items | Asia/Pacific | | Europe | | Americas | | Middle E./Africa | | Total | |
|---|---|---|---|---|---|---|---|---|---|---|
| | n | % | n | % | n | % | n | % | n | % |
| **Tourists' education** | | | | | | | | | | |
| Small groups of tourists | 19 | 82.6 | 16 | 94.1 | 7 | 100 | 3 | 100 | 45 | 90.0 |
| Inclusivity-contacts with locals of diverse backgrounds | 21 | 91.3 | 10 | 58.8 | 5 | 71.4 | 2 | 66.7 | 38 | 76.0 |
| Meeting local people and learning from them | 12 | 52.2 | 7 | 41.2 | 2 | 28.6 | 0 | 0.0 | 21 | 42.0 |
| Poverty reductions measures-buying from small local stores/markets | 21 | 91.3 | 12 | 70.6 | 4 | 57.1 | 2 | 66.7 | 39 | 78.0 |
| **Tourists' satisfaction-Happy tourists** | 18 | 78.3 | 13 | 76.5 | 6 | 85.7 | 2 | 66.7 | 39 | 78.0 |

The most common sustainable tourist behavior featured in the commercial discourse was travelling in small groups (90%). More than three-quarters of the videos included travelers shopping at small local stores/markets hence contributing to poverty reduction and economic equity (78%) or engaging in contacts with locals of diverse backgrounds as a mark of inclusivity (76%). Vectors of inter-influences, the scenes with tourists actively learning from locals, were promoted in less than half of the videos (42%) and nonexistent in the case of Middle east/African tourism advertisements. Although the regions manifested differences in the frequencies of the educational items, the Chi-square analyses revealed they were not statistically significant ($p > 0.050$).

Further, the results showed that 78% of the total videos showed joyful tourists engaged in various activities. An inter-regional comparison placed the American tourism advertisements at the top (85.7%), followed in order by the Asian/Pacific videos (78.3%), European (76.5%) and Middle East/African (66.7%).

## 5. Discussion and Conclusions

According to the existing literature, there are no current measures specifically designed to assess the integration of the sustainable tourism concept into the projected destination image. Consequently, the current study contributes to the existing literature on destination image formation by proposing a content analysis grid which can be used to capture and evaluate the sustainable tourism dimensions as portrayed in advertising videos of cities or even countries. Moreover, it can serve as a starting point when analyzing the sustainability features in communication activities, regardless of the industry or the format of the communication material. At the same time, the paper enriches the theoretical knowledge by providing insights into the destination image formation process from the tourism stakeholders' perspective [104,119,121].

Although the major informative role of social media in the travel decision-making process and in relation to the destination image formation has been extensively docu-

mented, and has even changed the marketing communication landscape for the tourism stakeholders [100,101,120,124], the current analysis revealed that seven of the most visited cities in the world did not have presentation videos on the DMOs' official YouTube and Facebook pages.

When projecting sustainable-oriented communication strategies, the tourism stakeholders need to follow and go beyond the principles of "green advertising" to encompass not only the environmental aspects [158] but also the socio-cultural and economic dimensions; thus, it can be said they are heading towards a new paradigm—a "sustainable advertising". While the literature on "green advertising" communication strategies does not provide clear and comprehensive requirements and practices, it suggests an approach borrowed from the general advertising setting based on strong argument quality [159]. Particularizing the communication actions in the case of DMOs, the explicit use of specific sustainable tourism-related terminology in the advertising discourse can increase the persuasive power, effectiveness and relevance of the information/message, contributing to the content of a strong argument [159]. The synergy between a highly credible information source (DMOs) and high-quality information, manifested through strong argument quality in the advertising discourse, facilitates the projection of a sustainable destination image and emphasizes its persuasive effect on tourists' attitudes and behavioral intentions [159,160]. However, such specific terminology that emphasizes the sustainable approach in communication is lacking in all the tourism videos analyzed.

In a similar manner, the quantity, variety, complexity, and relevance of visual information (images and video frames depicted in advertisements) covering the economic, socio-cultural, and environmental dimensions of sustainable tourism, can considerably shape the argument quality. Aiming to provide a more comprehensive perspective on the depiction of tourism sustainability concepts in the advertisements, our approach to the discussion groups answers the aforementioned research questions (general integration of the sustainable tourism elements into the projected image of destinations, government/DMOs' initiatives for sustainable tourism development, promotion of sustainable tourist behavior).

To unveil the environmental/natural dimension of sustainable tourism, the advertising discourse included frames featuring a wide variety of natural elements such as a clean environment, urban parks/green spaces and natural areas outside the city, mild/pleasant weather, natural attractions, tourist outdoor activities (walking, kayaking, biking, etc.), lack of crowds, reduced, traffic. Yet, despite having a wide representation among videos, these elements were not revealed from a sustainable perspective. For instance, the advertisements containing frames with natural attractions such as seas, mountains, secular forests, and wild areas do not provide mentions related to a proper management of ecosystems, protection of natural areas, and sustainable use of natural resources [5,42,72,73,82]. Previous research suggests that tourists prefer nature sounds [161] that enhance their sense of involvement and immersion [162], yet they rarely were used on the video soundtrack. As a special remark, there were videos that displayed several nature harming activities and effects, including energy waste due to intense public building lighting, heavy traffic, crowds and implied water, air, and sound pollution. Although this approach might create more realistic expectations, it can negatively affect the projected destination image and discourage the visit [57]. The government environmental initiatives linked to sustainable development were mainly displayed through footage including electric transport vehicles and biodiversity preservation, whereas the green buildings and renewable energy were rarely showcased. The sustainable tourist behavior projected in most videos and associated with the environmental dimension was small group travel, which is reasoned to be a measure that helps reduce the over-tourism problem.

The socio-cultural dimension of sustainable tourism was largely depicted in the commercial discourse through footage exposing historical and cultural assets, socio-cultural heritage preservation and activities, inclusivity, and diversity. However, the videos displayed only a few socio-cultural elements from a sustainable point of view, without explicitly emphasizing the preservation of diversity, traditional values, and local integrity, and if the

social inclusion and cohesion, inter and intra-generational equity and human development were fostered. In terms of assets, the cultural and historical buildings, museums/theatres, and art galleries were prevailing in the advertisements, whereas the socio-cultural heritage preservation was depicted mainly through frames that included scenes from the local community's way of life, traditional food/folk art/clothing, crafts. The most promoted activities through which tourists could discover the socio-cultural heritage were the interactions with local people, visits to cultural/historical attractions, attendances at opera/theatre performances, and consumption of traditional products. In fewer instances, the authentic tourism experience was enhanced through appreciation of dances/music and visits to villages and traditional houses. Notably, less than half of the tourism advertisements displayed frames with traditional music and local language, despite being important assets of intangible cultural heritage [163]. The socio-cultural government initiatives connected to sustainable development were mostly depicted through footage implying the idea of tolerance and understanding towards diversity, respect for local food producers and local food as agents in the conservation of cultural heritage [42,53,71,72]. The responsible tourist behavior linked to the socio-cultural dimension and projected in advertisements through the explicit display of travelers, refers to interactions with locals of diverse backgrounds as an indicator of inclusivity and tourists actively learning from locals.

The economic dimension of sustainable tourism was depicted in the commercial discourse through frames featuring various economic elements like locally owned businesses, a good transportation system and infrastructure, shopping and accommodation units, and nightlife. Nevertheless, the footage did not explicitly expose the impact of tourism-related activities on economic growth and regional development. The stringent matter of safety with impact on destination image was covered in all videos [111,164], yet the smart city solutions were seldom displayed regardless of the synergies between smartness and sustainable development [71,165,166]. The government economic initiatives affiliated with sustainable development were conveyed through scenes presenting small premises with friendly sellers aimed at tourists and local micro-business with diverse ownership, reasoned to be contributors to poverty reduction and economic equity [47]. In spite of their importance for travelers, the traditional accommodation units [167] and the local partnerships (information centers, local tourism associations) were hardly shown [168]. The tourist behavior projected in advertisements with an explicit presence of travelers refers to purchases from small local stores/markets, thus aligning to the promoted government initiatives.

In the global context that drives tourism towards a sustainable approach [13,169] and given that destination image influences all stages of the travel decision-making process [104], the intention to visit a sustainable destination [112–114] and to adopt a sustainable tourist behavior [115], the tourism stakeholders (as DMOs) must integrate the sustainable tourism dimensions in their communication activities aimed at destination image projection. In view of that, and based on the findings of the study, we propose a set of recommendations for marketing specialists in the field. Primarily, all DMOs must incorporate social media platforms in their communication strategies and create and share promotional videos of tourist destinations. Further, including specific sustainable tourism-related terminology (verbal, textual) in the advertising discourse enriches the sustainable approach in communication, whilst adding natural ambient sounds and traditional music could enhance the tourist experience. The tourism videos must broaden their focus by adding frames with overall rare depictions such as local healthy products, visits to traditional houses, local partnerships, smart city solutions, renewable energy, and green buildings. Besides, the government initiatives towards sustainability commitment and tourist education through explicit travelers' depiction (induced behaviors) need to be explored more. However, an indirect exposure of the sustainability-related elements is not reasoned to be sufficient in building a sustainable destination image and needs to be accompanied by an explicit display in a sustainable manner. Considering the scarce frames included in the advertising discourse, several recommendations emerge by regions: (a) Asian/Pacific advertisements—to include footage featuring eco-friendly transportation

means, (b) European videos—to raise the number of frames that depict natural resources, traditional music/arts/goods, local scenes of life, tourist accommodation, (c) American videos—to increase the focus on the socio-cultural dimension through frames depicting cultural preservation, traditional music/crafts/food/products, visits to cultural attractions and theatres/museums/art galleries and on sustainability in terms of transportation means, (d) the Middle East/African advertisements—to extend the footage displaying scenes with visits to cultural attractions, theatres/museums/art galleries, traditional crafts/music, and electric transportation.

The current research has several limitations that could represent a starting point for future research directions. The selected advertisements came from DMO's accounts on two social media platforms: YouTube and Facebook. Consequently, the sampling process took into account a few missing promotional videos for highly visited cities, whereas the findings cannot be generalized across all social media channels. In future studies, the sampling can extend to comprise all online communication tools, both in terms of social media channels and official DMOs websites. Although the purposing sampling was correctly undertaken and the sample size was consistent for qualitative research, the sampling structure was unbalanced (especially when considering the 3 cities in the Middle East/Africa) and reflected in the inter-regional comparisons. Thus, future scholars could develop a sampling method and/or size that provides a more balanced sample structure if inter-regional comparison is wanted. Given the exploratory nature of this study, the proposed evaluation grid should be seen as an element of novelty in research with theoretical implications but also as a limit, because the literature review provided only a few solid measures on tourism sustainability to include in the inquiry. This calls for future research to explore and provide a richer understanding. In terms of data analysis, a future in-depth approach can extend to different functions provided by specialized VCA software/technology.

**Author Contributions:** Conceptualization, M.F.B. and R.C.; methodology, M.F.B. and R.C.; validation, R.C. and M.F.B.; formal analysis, M.F.B.; investigation, M.F.B., R.C., L.M.S. and A.M.D.; resources M.F.B.; datacuration, M.F.B.; writing—original draft preparation, M.F.B., R.C., L.M.S. and A.M.D.; writing—review and editing, R.C. and M.F.B.; visualization, M.F.B. and R.C.; supervision, R.C.; project administration, M.F.B.; funding acquisition, M.F.B. All authors have read and agreed to the published version of the manuscript.

**Funding:** This research was founded by Babeș-Bolyai University grant Fondul de dezvoltare UBB 2021, project GS-UBB-FSEGA-bacilamihaiflorin.

**Institutional Review Board Statement:** Not applicable.

**Informed Consent Statement:** Not applicable.

**Data Availability Statement:** Not applicable.

**Acknowledgments:** Not applicable.

**Conflicts of Interest:** The authors declare no conflict of interest.

## Appendix A

**Table A1.** Content analysis grid.

| Categories | Sub-Categories & References | Operational Definitions |
|---|---|---|
| Facets of projected destination image | [57,79,134,170] | Natural areas destinations-sea/mountains |
| | | Preserved nature attractions-secular forests/wild areas |
| | | Urban areas |
| | | Cultural destinations-historic city/ruins |
| | | Cultural destinations-cultural heritage |

**Table A1.** *Cont.*

| Categories | Sub-Categories & References | Operational Definitions |
|---|---|---|
| Evidence on sustainable tourism dimensions | | Natural attractions-sea, mountain, river |
| | | Natural areas outside the city |
| | | Parks and green spaces in the city |
| | Natural dimension [3,9,25,45,57,79,80,82,87,134,171–173] | Mild/nice weather |
| | | Clean environment |
| | Natural/ environmental dimension | Reduced traffic |
| | | Lack of crowds |
| | | Natural ambient sound |
| | | Outdoor natural strolls/trekking |
| | | Intense traffic |
| | | Crowds |
| | Nature harming activities and effects [3,5,11,12,42,73,79,85,87,88,174,175] | Implied water and air pollution |
| | | Implied sound pollution |
| | | Intense public lighting |
| | | Cultural attractions |
| | | Preserved historical buildings |
| | | UNESCO awards/distinctions |
| | | Museums, theatres, art galleries |
| | | Festival/event |
| | | Crafts/handmade art |
| | | Local traditions and cultural identity-including folk art and clothing |
| | | Dance/ music |
| | | Traditional food |
| | | Green label/ healthy products |
| | | Cultural preservation |
| | Socio-cultural dimension [2,3,9,23,54,63–65,68,70,76,78,114,117,134,161–163,167,171,173,176] | Scenes from the local community's way of life |
| | | Activities–listening to music, dancing |
| | | Activities–visits to cultural/historical attractions |
| | | Activities–attending opera or theatre performances or art exhibitions |
| | | Activities–interaction with local people |
| | | Activities–visits to rural areas/villages/ traditional houses |
| | | Activities–consuming traditional products |
| | | Culinary experiences in a traditional restaurant |
| | | Culinary experiences in a non-traditional restaurant |
| | | Friendly locals |
| | | Ambient sound–traditional music/local language |
| | | Strong economy |
| | | Safe cities-the absence of underdeveloped /high-crime areas/ neighborhoods |
| | | Good infrastructure |
| | Economic dimension [1,3,12,53,54,65,70,71,73,82,87,134,164–166,171,173,177–179] | Smart city |
| | | Good transport system |
| | | Hotels/resorts |
| | | Local businesses |
| | | Activities–shopping |
| | | Nightlife |

**Table A1.** *Cont.*

| Categories | Sub-Categories & References | | Operational Definitions |
|---|---|---|---|
| Government initiatives related to sustainable tourism development | Natural/environmental dimension [9,12,54,73,79,83,84,87,134,172,173,176,177] | | Focus on natural resources |
| | | | Biodiversity preservation–plants |
| | | | Biodiversity preservation–animals |
| | | | Renewable energy |
| | | | Green buildings |
| | | | Electric transport vehicles-train, tram, metro |
| | Socio-cultural dimension [12,17,53,54,64,65,72,78,114,117,134,173] | | Respect for the local food producers |
| | | | Promoting traditional/ eco-friendly food- local food markets/street food |
| | | | Tolerance and understanding-gender, sexuality, race and age diversity |
| | Economic dimension [1–3,5,12,47,54,71,73,74,78,114,134,168,171] | | Traditional accommodation units |
| | | | Small local businesses own by diverse locals |
| | | | Service quality–friendly sellers |
| | | | Service quality–small physical premises |
| | | | Local partnerships-info centres and local tourism associations |
| Explicit presence of tourists | Tourists' education | Natural/ environmental dimension [9,79,134] | Small groups of tourists |
| | | Socio-cultural dimension [3,9,12,25,37,47,54,64,74,76,78,88,101,114,134] | Inclusivity-contacts with locals of diverse backgrounds |
| | | | Meeting local people and learning from them |
| | | | Poverty reductions measures-buying from small local stores/markets |
| | Tourists' satisfaction [3,45,50–52,54,63,64,108,109,132,134,176] | | Happy tourists |
| Sustainable tourism-related terms | [1,2,4–6,8,9,12,23,25,35,39,40,42,43,53,70–76,82,83,105,107,112,158] | | Sustainability, sustainable development/tourism, equitable/responsible/eco/green tourism, eco-friendly, nature/biodiversity/ ecosystem preservation, cultural preservation, inclusivity, fair/equitable income distribution, poverty alleviation, responsible travel/ behaviour |

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
