# Peer review of "Content Analysis on Sustainability Dimensions in DMOs’ Social Media Videos Advertising the World’s Most Visited Cities"

_sustainability, doi:10.3390/su141912333_

Round 1

Reviewer 1 Report

Thank you for the opportunity of reading and reviewing your manuscript. It addresses an interesting topic which falls under the journal's scope. The paper reviews the relevant references, and the methodology used for investigation is appropriate. The manuscript is well written and there are relevant research results presented. I have several suggestions / comments to increase readability:

1.it would be beneficial to construct research hypotheses based on the literature instead the more general research questions, maybe you can work on it

2.please rename section 4 from Results to Results and discussion and include more in-depth discussion of your findings

3.please refer in the final section to the implications both theoretical and managerial, of your results.

Good luck!

Reviewer 2 Report

First of all, I want to thank the authors for producing a very interesting research paper. It is a research of social media communication video content analysis of the most visited cities in the world. It is a quite original and interesting topic in the field of tourism destination promotion and virtual tour. I have some thoughts that I would like to share with the authors that they are recommended to take seriously in order to present an improved version of this research paper.

More detail please see attachment 

Reviewer 3 Report

Dear Authors,

First and foremost, please accept my gratitude for your time and effort in writing this manuscript. In my opinion, this is a well-structured paper. It is strongly supported by relevant literature, and the reliability concerns have been adequately addressed in the Methodology section.  Moreover, I agree that the best way to study projected destination image is through content analysis. Purposive sampling is also well justified. However, a few adjustments would improve the article’s quality.

1. Please provide the full form of all abbreviations that appear in the article. After all, DMO is not a widely spread acronym.

2. The abstract should contain one or two sentences explaining the article's originality and methodology.

3. Please, spend more time in the Methodology section explaining how exactly the video material was selected and what were your criteria to establish whether the selected video material was posted by DMOs or others.

4. I would suggest you create another table, reporting all the significant chi-square test results together. Otherwise, it is difficult to navigate between different sub-sections in the attempt to remember them all.

5. Finally, try to emphasize in conclusions how exactly you have answered the three research questions, otherwise, including them in the first place seems a bit useless.

Best regards

Round 2

Reviewer 2 Report

I would like to thank the authors for revising the manuscript based on the comments of the first review. However, there are still several omissions from the comments of the first review, which I kindly point out to the authors for a possible second, more carefully prepared version of the manuscript, that should be taken seriously:

1. The title of the manuscript is still not well supported by the results. The question remains how DMOs marketing communication strategies relate to sustainability dimensions.

2. The literature review still refers to sustainable development and destination image while the purpose of the research aims to evaluate sustainability dimensions in DMOs social media communication videos. Based on the objectives of the research as formulated the literature review should focus on the dimensions of sustainability and how they relate to social media communication videos DMOs of tourist destinations.

3. The manuscript still poses three research questions (RQ1, RQ2, RQ3) without addressing them separately in the discussion section.

4. It is still not clearly understood what the research findings are in relation to the research questions.
